# The Parent's Chronotype and Child's Sleeping Quality in Association with Relationship Satisfaction

**Cristian Ricci [1,*], Zaida Parra-Robledo [2], Dietrich Rothenbacher [3], Juan Francisco Díaz-Morales [2] and Jon Genuneit [1,3]**

1. Pediatric Epidemiology, Department of Pediatrics, Medical Faculty, Leipzig University, 04103 Leipzig, Germany; Jon.Genuneit@medizin.uni-leipzig.de
2. Individual Differences, Work and Social Psychology Department, Universidad Complutense de Madrid, 28223 Madrid, Spain; zparra@ucm.es (Z.P.-R.); jfdiazmo@ucm.es (J.F.D.-M.)
3. Institute of Epidemiology and Medical Biometry, Ulm University, 89081 Ulm, Germany; Dietrich.Rothenbacher@uni-ulm.de
* Correspondence: Cristian.Ricci@medizin.uni-leipzig.de

**Abstract:** The prospective Ulm-SPATZ study was investigated to assess the role of child sleeping quality between 4 to 6 years of age in affecting a partner's sleeping and relationship satisfaction within a couple. The study was conducted using a triadic approach in which the child was included in the Actor-Partner-Interdependence Model (APIM). Sleeping quality of the child was determined by using the German version of the children's sleep habits questionnaire, sleeping features of the parents were assessed by using the Munich chronotype questionnaire, and the partner relationship assessment was performed by employing the German version of the parenting stress index questionnaire. In 211 German triads, we observed that sleeping characteristics and partner relationship scores at different child ages are consistent for both men and women. Higher and statistically significant sleep duration, time spent in bed, the midpoint of sleep, time getting out of bed, and sleep onset in women compared to men during the working days were observed. The APIM analyses showed a significant direct effect of child sleep quality on the partner relationship satisfaction. In women, a mediated effect of child sleep quality acted through sleep duration and time spent in bed on the partner relationship satisfaction score during both free and working days. In men, low child sleep quality was found to be associated with increased sleep onset during both free and working days. Child sleep quality influences relationship satisfaction mostly in mothers, likely because of their higher involvement in childcare during working days. Distress in the couple could be counteracted by a major involvement of the fathers in child management.

**Keywords:** sleeping quality; chronotype; SPATZ study; family

## 1. Introduction

The chronotype is the behavioral manifestation of underlying circadian rhythms that govern the propensity of an individual to sleep at a particular time during the 24-hour period. Individuals have relevant flexibility in the timing of their sleep period, meaning two divergent chronotype phenotypes can be observed, along with their intermediate features [1]. On one hand, morning-oriented persons are those who prefer to get up early and reach physiological characteristics such as hormone levels and their maximum cognitive performance and well-being during the morning. On the other hand, evening-oriented persons are those who prefer to get up late and stay up longer. Generally, evening-oriented persons have a shift of peak physiological characteristics towards later hours and experience both better cognitive performance and well-being during the afternoon or in the evening.

The chronotype likely affects relationship satisfaction and well-being of individuals involved in a romantic relationship. Firstly, couples tend to have similar chronotype and sleeping habits [2]. Secondly, the chronotype similarity positively affects the partnership relationship between the two people in the couple [3]. Furthermore, the association between chronotype similarity and relationship satisfaction operates as a function of the relationship stage. In early couples, with a relationship length of approximately four years, higher chronotype similarity is related to better relationship satisfaction, whereas in more mature couples with a relationship duration of up to approximately 25 years, the chronotype similarity does not affect the partnership relation [4–6].

Sleep problems and relationship problems tend to co-occur, particularly during times of significant life events or transitions, such as the birth of a child [7]. In accordance, literature reported that in different-sex couples, relationship satisfaction decreases after the birth of a child [8]. In particular, likely because of the child crying and other sleep-related problems, the first years of parenthood appear as the most critical period for the relationship satisfaction [9]. Afterwards, when the child reaches around 7 years of age, the relationship satisfaction restores, possibly because the child becomes more independent and his/her sleep-related problems reduce [10].

According to accepted theories, the couple and their child might be interpreted as a single system in which all elements influence each other [11]. So, the theory cannot be excluded that the parents' chronotype, relationship stage, and the child's sleep quality are inter-correlated and likely influence the couple relationship.

The objective of the present study was to investigate to what extent the child's sleep quality influences the parents' sleep and their couple relationship satisfaction. The present work had 3 specific aims. Firstly, we aimed to determine how the sleep quality of the child directly influences the partner relationship satisfaction. Secondly, we evaluated if an indirect effect of the sleep quality of the child on the partner relationship satisfaction subsists through the sleep of the parents. Thirdly, we aimed to determine how the chronotype and sleep duration of a person engaged in the couple influences his/her own relationship satisfaction (actor effect) or the partner's relationship satisfaction (partner effect). Moreover, the present study also investigated the relation between the sleep quality of the child and the partner relationship satisfaction over three waves of the SPATZ study, while also evaluating the effects given by the age of the child. Finally, sleep duration and the midpoint of sleep were assessed during working and free days, giving us the opportunity to evaluate how a working routine influenced the relation between the sleep quality of the child and the partner relationship satisfaction. To achieve these aims, data on parents and children from the Ulm SPATZ Health Study were analyzed as a whole using a comprehensive triadic approach based on the Actor-Partner-Interdependence Model [12].

## 2. Results

### 2.1. Participant Characteristics

After deleting records with missing values for sleeping characteristics, partner relationship scores, child sleep quality score, and duplicated records for twins (one twin was randomly selected in each case), 211 triads remained in the analytical dataset. The median baseline age of men was 37 years (range 28 to 59), the median age of women was 36 years (range 27 to 46), and the median age difference between men and women engaged in a relationship was 2 years (a range of −6 to 18). A similar number of men and women had a higher education degree (70–75%).

### 2.2. Longitudinal Assessment of Chronotype and Partner Relationship Score

Sleeping characteristics and partner relationship scores at different child ages appeared to be consistent for both men and women (Table 1). Still, differences were reported within couples, with higher and statistically significant differences in sleep duration, time spent in bed, the midpoint of sleep, time getting out of bed, and sleep onset in women compared to men during working days (men reported a longer sleep duration and more time spent in bed, an earlier midpoint of sleep, getting out

of bed earlier, and an earlier sleep onset). When looking at within couple differences during free days, we found higher statistically significant values for all sleeping features, but not for time spent in bed, in men compared to in women (Table 2). Fewer within couple differences were observed when the age of the child was 6 years old, compared to the earlier time points.

Within couples, we reported a negative correlation between the partner relationship score and the midpoint of sleep, the time of getting out of bed, and the sleep onset during working and free days and in the first two time points of the study. This correlation was somehow weaker when the age of the child was six years old. Notably, the within couple correlation between midpoint of sleep, time of getting out of bed, and sleep onset appeared to be stronger during free days, irrespective of the time point. Moreover, we observed moderate to high within couple correlations of the partner relationship score; these correlations were comparable at different ages of the child and different time points during both free and working days. Finally, we reported a moderate to strong negative correlation between fathers' MSN, GUN, and SON and mothers' TBT and SD at wave 6 during working days, but not on free days (Figure 1). When looking at the correlation structure over time, we observed a good consistency between the same variable registered at different time points during both free and working days. However, the time correlation pattern to be sparse and inconsistent when looking at the off-diagonal elements. In particular, we observed differences between men and women in the correlation structure of MSN, GUN, and SON. These items appeared to be more highly correlated in men than women and were consistent over the observational time. Furthermore, these correlations appeared to be stronger during working days compared to during free days (Figures 2 and 3).

**Table 1.** Medians and 10th percentile–90th percentile ranges of relationship scores and sleeping features of 221 couples when their children are at specific ages (in years) while participating in the SPATZ study.

| | Men | | | Women | | |
|---|---|---|---|---|---|---|
| | **Child's Age: 4** | **Child's Age: 5** | **Child's Age: 6** | **Child's Age: 4** | **Child's Age: 5** | **Child's Age: 6** |
| PB | 6 (2) | 6 (3) | 5 (2) | 7 (3) | 6 (3) | 7 (2) |
| Working days | | | | | | |
| SD (HH:MM) | 06:20 (01:30) | 07:00 (01:10) | 07:20 (01:40) | 07:00 (01:50) | 07:00 (01:30) | 07:15 (01:20) |
| TBT (HH:MM) | 06:50 (01:30) | 07:40 (00:35) | 07:45 (00:20) | 08:15 (00:55) | 08:00 (01:00) | 07:50 (01:05) |
| MSN (HH:MM) | 02:30 (00:30) | 02:40 (00:45) | 02:45 (00:30) | 02:30 (00:30) | 02:45 (00:50) | 02:15 (00:35) |
| GUN (HH:MM) | 05:20 (00:35) | 05:10 (00:35) | 05:10 (00:50) | 05:30 (00:45) | 05:30 (01:00) | 05:25 (01:10) |
| SON (HH:MM) | 21:30 (01:05) | 21:30 (00:55) | 21:30 (00:55) | 21:15 (01:10) | 21:15 (00:50) | 21:30 (00:55) |
| Free days | | | | | | |
| SD (HH:MM) | 07:30 (01:00) | 07:30 (01:50) | 07:30 (01:10) | 07:30 (01:00) | 07:40 (01:00) | 07:50 (00:50) |
| TBT (HH:MM) | 08:00 (01:00) | 07:40 (00:50) | 07:50 (01:00) | 08:00 (00:50) | 08:00 (00:55) | 08:20 (01:00) |
| MSN (HH:MM) | 03:30 (00:30) | 03:30 (01:45) | 03:00 (01:15) | 02:35 (01:25) | 02:40 (01:30) | 03:00 (01:20) |
| GUN (HH:MM) | 06:30 (01:00) | 06:50 (01:55) | 06:20 (01:35) | 06:10 (01:50) | 06:15 (01:50) | 06:15 (01:48) |
| SON (HH:MM) | 22:00 (01:10) | 22:30 (01:00) | 22:30 (01:30) | 22:00 (01:30) | 22:30 (01:00) | 22:15 (00:50) |

Notes. PB: Relationship satisfaction score, SD: Sleep duration, TBT: Time spent in bed, MSN: Midpoint of sleep, GUN: Time getting out of bed, SON: Sleep onset.

**Table 2.** Medians and 10th percentile–90th percentile ranges of women-men partner relationship score and sleeping differences in 221 couples participating in the SPATZ study.

|  | Child's Age: 4 Years | | Child's Age: 5 Years | | Child's Age: 6 Years | |
|---|---|---|---|---|---|---|
|  | p50 (p10; p90) | $P_{binomial}$ | p50 (p10; p90) | $P_{binomial}$ | p50 (p10; p90) | $P_{binomial}$ |
| PB | 1 (−1; 3) | <0.001 | 0 (−1; 3) | <0.001 | 0 (−1; 4) | 0.346 |
| Working days |  |  |  |  |  |  |
| SD (minutes) | 50 (−30; 90) | <0.001 | 0 (−90; 90) | 0.999 | 15 (−105; 105) | <0.001 |
| TBT (minutes) | 60 (−30; 90) | <0.001 | 30 (−30; 90) | <0.001 | 30 (−75; 120) | <0.001 |
| MSN (minutes) | −2 (−75; 35) | 0.501 | 30 (−70; 45) | <0.001 | 20 (−110; 70) | <0.001 |
| GUN (minutes) | 15 (−30; 60) | <0.001 | 5 (−45; 75) | 0.419 | 15 (−60; 50) | <0.001 |
| SON (minutes) | −15 (−120; 15) | <0.001 | −15 (−90; 30 | <0.001 | 0 (−120; 90) | 0.788 |
| Free days |  |  |  |  |  |  |
| SD (minutes) | 1 (−40; 90) | 0.999 | 10 (−90; 90) | 0.501 | 15 (−180; 65) | <0.001 |
| TBT (minutes) | 0 (−30; 89) | 0.501 | 30 (−30; 90) | <0.001 | 30 (−150; 60) | <0.001 |
| MSN (minutes) | −30 (−60; 15) | <0.001 | −15 (−65; 20) | <0.001 | −3 (−90; 30) | 0.139 |
| GUN (minutes) | −15 (−60; 30) | <0.001 | −30 (−60; 30) | <0.001 | 0 (−90; 30) | 0.139 |
| SON (minutes) | −30 (−120; 0) | <0.001 | −31 (−90; 0) | <0.001 | −15 (−120; 90) | <0.001 |

Notes. p50: Median of women-men difference, (p10; p90): range 10th percentile–90th percentile of women-men difference, $P_{binomial}$: *p* value for the exact binomial test comparing partners, PB: Relationship satisfaction score, SD: Sleep duration, TBT: Time spent in bed, MSN: Midpoint of sleep, GUN: Time getting out of bed, SON: Sleep onset.

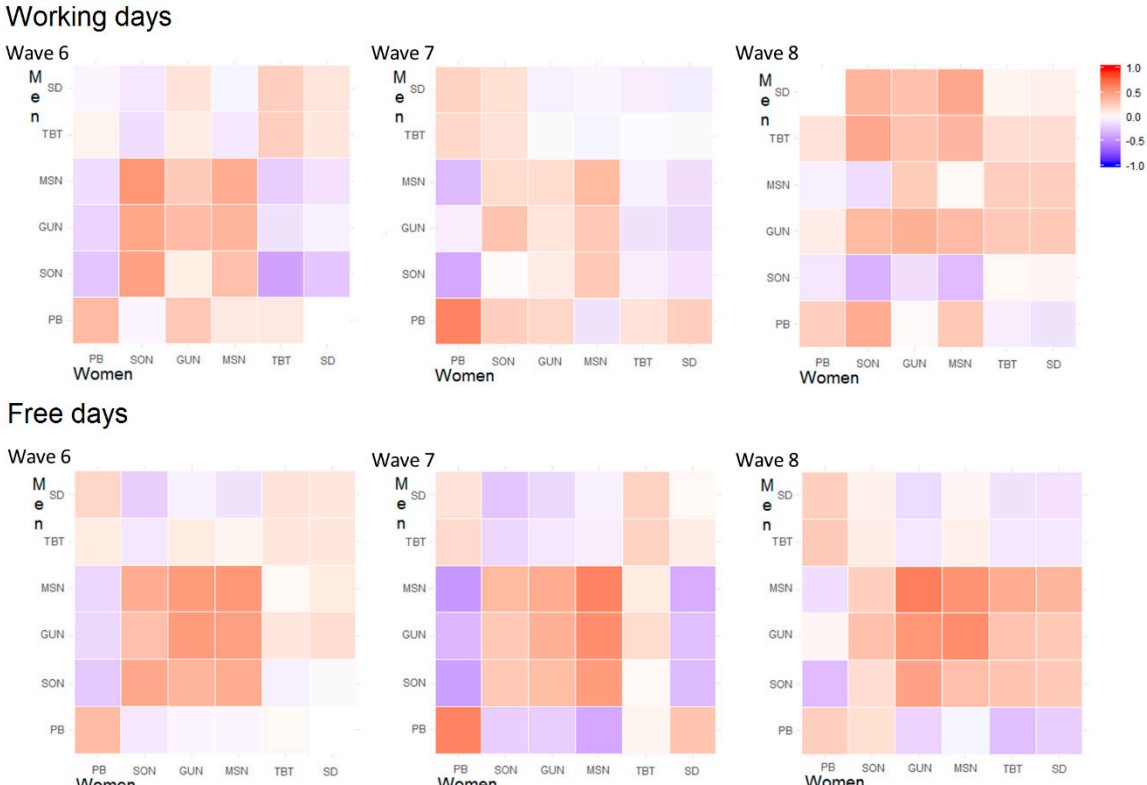

**Figure 1.** Heat maps of Pearson correlation coefficients of sleeping features and relationship satisfaction scores at different time points for women vs. men. Note: SCC: Child sleeping quality score, PB: Partner relationship score, SD: Sleep duration, TBT: Time spent in bed, MSN: Middle sleep, GUN: Time getting out of bed, SON: Sleep onset.

### 2.3. APIM Analysis

The APIM model analyses showed a significant direct effect of child sleep quality on the partner relationship satisfaction. In women, we also observed a mediated effect of child sleep quality on the partner relationship satisfaction score. This mediation acted through sleep duration and time spent in bed. This result was observed during both free and working days and was confirmed when looking at the midpoint of sleep on working days but not on free days.

In men, it appears that low child sleep quality is associated with increased sleep onset during both free and working days. Moreover, during free days we observed a significant mediated effect on the partner relationship satisfaction score. This mediation was realized through an indirect effect via the midpoint of sleep.

When looking at how the partners' sleeping features may have affected partner relationship satisfaction scores, we observed a significant partner effect of women on the partner relationship satisfaction score of men for time spent in bed, the midpoint of sleep, the sleep duration, and the sleep onset during both free and working days. A statistically significant actor effect on the partner relationship score of women was observed for all sleeping features during free and working days, apart from the time of getting out of bed during the free days. In men, a statistically significant actor effect on the partner relationship satisfaction score was observed for the midpoint of sleep during free and working days and for time of getting out of bed during free and working days. All the APIM analyses considering different sleeping features showed similar model fitting with BIC values ranging between 7630 and 8440.

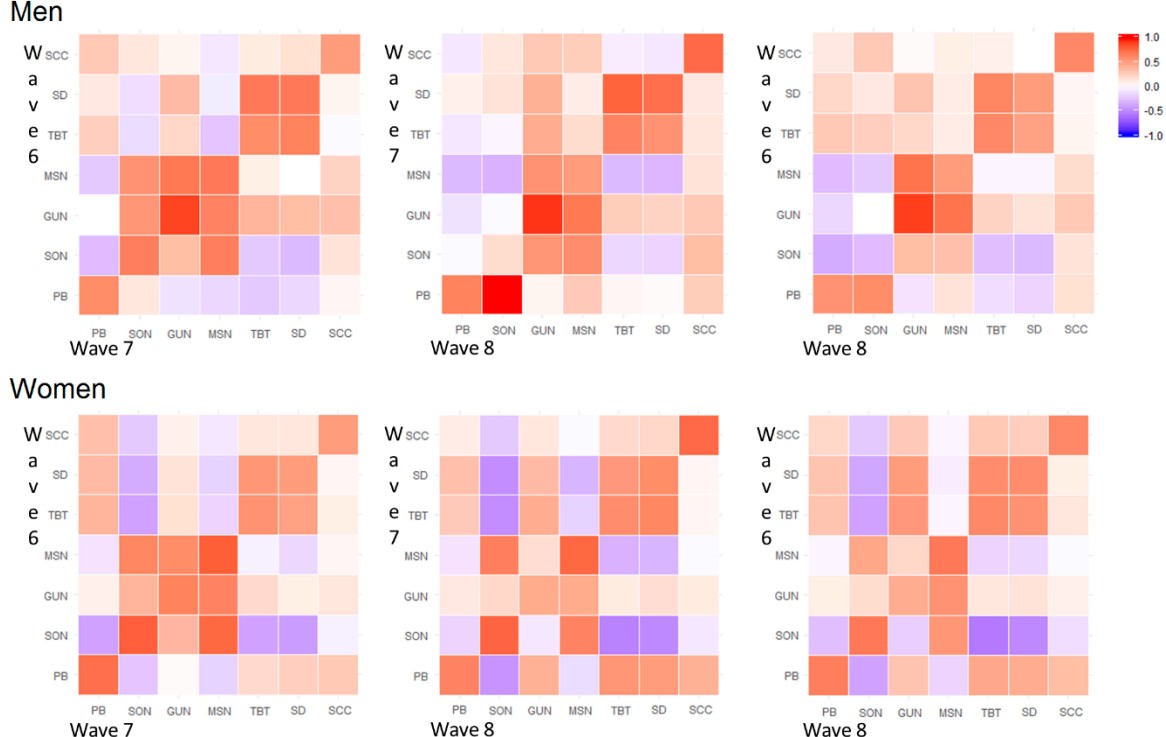

**Figure 2.** Heat maps of Pearson correlation coefficients of sleeping features, child sleep quality scores, and relationship satisfaction scores for different time points during working days. Note: SCC: Child sleeping quality score, PB: Partner relationship score, SD: Sleep duration, TBT: Time spent in bed, MSN: Middle sleep, GUN: Time getting out of bed, SON: Sleep onset.

For example, after considering the APIM analysis performed on sleep duration during working days, we reported a moderate to strong significant association of sleep quality of the child on the partnership score of the mother (r = 0.31). This relation is mediated through the sleep duration of the mother, with a child being asleep found to be moderately associated with the sleeping of its mother (r = 0.25), while a strong association was also found of the sleep duration of the mother on her own partnership score (r = 0.51). On the other hand, we reported a moderate association of sleep quality of a child on the partnership score of the father. In fathers, this appears to act directly without a mediating role of the sleep duration. Finally, a medium association between the sleep duration of the mother and the partnership score of the father emerged (r = 0.24).

The current results were confirmed in sensitivity analyses further adjusted for occupation, body mass index, parity, smoking status, and alcohol use (Figure 4).

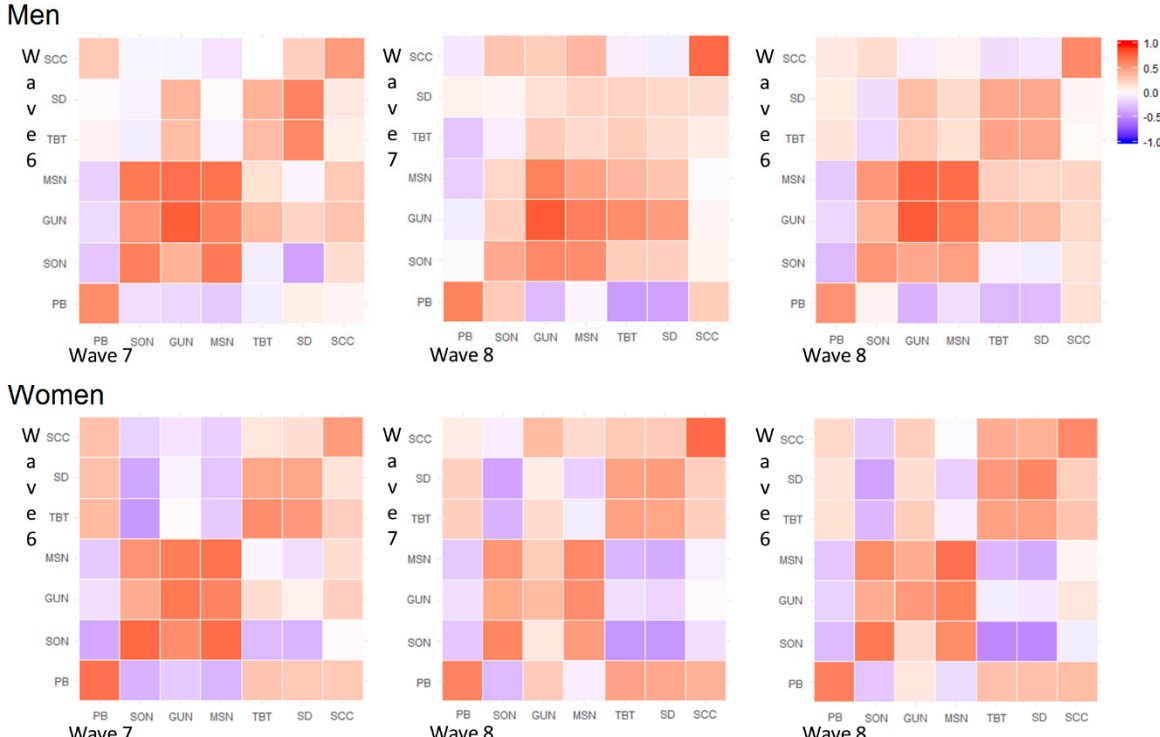

**Figure 3.** Heat maps of Pearson correlation coefficients of sleeping features, child sleep quality scores, and relationship satisfaction scores for different time points during free days. Note: SCC: Child sleeping quality score, PB: Partner relationship score, SD: Sleep duration, TBT: Time spent in bed, MSN: Middle sleep, GUN: Time getting out of bed, SON: Sleep onset.

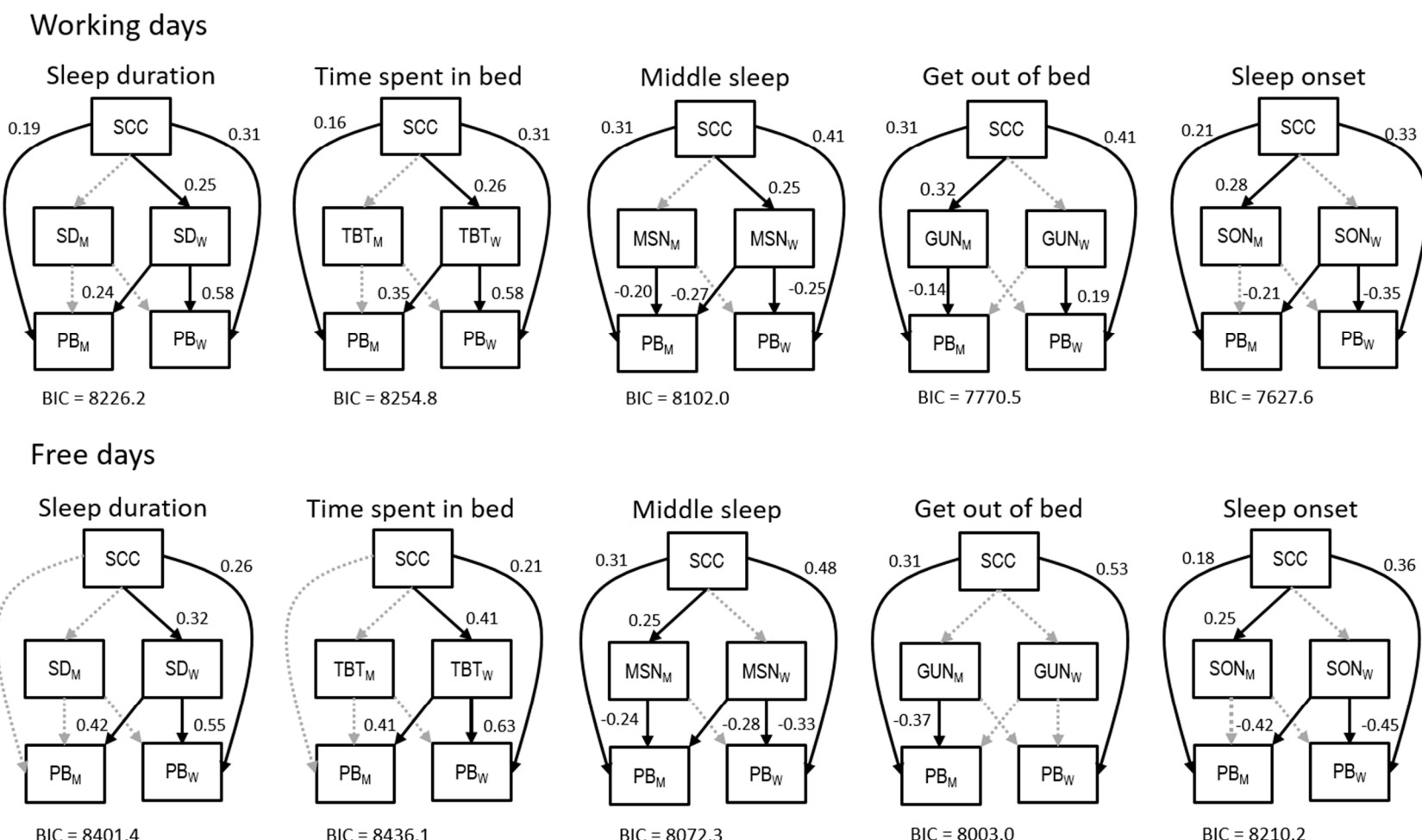

**Figure 4.** The Actor-Partner Interdependence model performed during free and working days. Note: SCC: Child sleeping quality score, PB: Partner relationship score, SD: Sleep duration, TBT: Time spent in bed, MSN: Middle sleep, GUN: Time getting out of bed, SON: Sleep onset.

## 3. Discussion

The present work including $n = 221$ triads from a birth cohort in Germany showed some statistically significant chronotype differences within couple. In particular, different patterns of chronotype differences within couple were observed during working and free days. More specifically, we reported that women have higher sleep duration, go to bed, and get out of bed later compared to their male partner during the working days. We also reported that this pattern is inverted during free days, where we also observed a longer sleep duration of both women and men. This could be interpreted as men waking up early for work during the week and recovering lost sleep during free days, a general pattern which is well acknowledged in the current scientific literature [13].

Moreover, we also observed that sleeping features are highly correlated within a couple. This correlation is stronger during when the couple's child is younger but dilutes at 6 years of age postpartum in our data, confirming that couples tend to have similar chronotype and sleeping habits in the earliest phases of their relationship, as already observed by other studies [3]. On one hand, within couple correlations of midpoint of sleep, time of getting out of bed and sleep onset are positive and particularly strong, especially during free days. On the other hand, the within couple correlations of sleep duration and time spent in bed are weak. Moreover, we here reported a negative correlation between chronotype proxies of the father (the midpoint of sleep, getting out of bed, and sleep onset) and sleep duration features of the mothers (the time spent in bed and sleep duration). This correlation appears to be stronger during the working days and when the child is younger. This result may point out that the father, more commonly the homeowner and the main source of family income, is less influenced by the child's sleep than the mother, who is likely to be more involved in the care of the child [14,15].

When looking at the relation between chronotype features and relationship satisfaction score, we observed that those chronotype features that are highly correlated within a couple (the midpoint of sleep, time of getting out of bed, and sleep onset) are also highly correlated with the relationship satisfaction score, showing that higher similarity in sleeping habits and chronotype may result in higher partnership satisfaction, as for other features [16]. This evidence was observed within the study wave but it also holds prospectively, with the child's age rising from four to five and from five to six years of age.

When looking at our APIM analyses, we observed that child sleep quality positively and directly affects the relationship satisfaction score. This effect is stronger in women compared to for their partner, likely because women are more involved in child care [17–19].

Moreover, an indirect effect of child sleep quality on the relationship satisfaction score was also observed in women. This indirect effect was mediated through the sleep duration, time spent in bed, and midpoint of sleep during working days and was also observed for sleep duration and time spent in the bed during free days. Notably, this indirect effect acts via two distinct aspects of the mother's sleep. Firstly, it acts via the midpoint of sleep, which is most likely an endogenous or biologically determined feature of the chronotype [20]. This results agree with other studies showing how a shift in the chronotype may results in a reduced psychological wellbeing that may lead to an impaired mood quality that could finally lead to the perception of a reduced partner relationship quality [21]. Secondly, it acts via sleep duration and the time spent in bed. This may follow a similar logic, since an adequate sleep quantity might have been related to impaired psychological and physical wellbeing [22].

In the fathers, child sleep quality has a direct effect, but not an indirect effect, on the relationship satisfaction score. Moreover, probably due to a lower involvement with childcare, the effect of child sleep quality on the relationship satisfaction score is lower in men with respect to their female partners, especially during working days. However, we also reported a statistically significant indirect effect of child sleep quality on relationship satisfaction on men. This effect was observed during working days for the time at which the individual gets out of bed and during free days for the midpoint of sleep. On one hand, men are likely to have to wake up early during working days, which can create distress in young fathers [23]. The possibility cannot be excluded that child sleep quality influences this aspect, leading to a negative effect on relationship satisfaction. On the other hand, during free days, fathers

have the time to recover from work and a negative effect of child sleep quality on the chronotype, here proxied by the midpoint of sleep, may increase the distress, leading to an effect on the relationship satisfaction score. Finally, on one hand, we reported a significant partner effect of almost all of women's sleeping features on men´s relationship satisfaction scores, excluding the time for getting out of bed. This effect was observed for both measures of the chronotype and for measures of sleep duration as well. Moreover, this effect was observed consistently, on both working and free days. On the other hand, the counterpoised partner effect of men on women was never observed for any of the sleep features investigated, either on working or on free days. This is novel evidence that could indicate specific coping mechanisms of women dealing with preschool children [24,25]. Moreover, gender differences in coping with distress were previously observed for other conditions such as infertility or chronic diseases [25–27].

## 4. Methods

### 4.1. The Ulm SPATZ Health Study

The Ulm SPATZ Health Study is an ongoing study based on 1006 newborns and their parents consecutively recruited during their hospital stay soon after delivery in the University Medical Center of Ulm, Southern Germany. Recruitment took place between April 2012 and May 2013. Yearly, on every birthday of the child, the families are contacted for a follow-up with separate self-administered questionnaires for each family member. So, the SPATZ study is a prospective birth cohort study with multiple measures. Records were excluded under the circumstances of: mothers having inadequate German language skills, outpatient childbirth, a maternal age of <18 years, postpartum transfer of the mother or the child to an intensive care unit, or a stillbirth. Participation in the study was completely voluntary and informed consent was obtained by all mothers and their partners. Ethical approval was obtained from the Ethics board of Ulm University (No. 311/11). The current study is based on three waves of the SPATZ study, namely waves 6, 7, and 8 when the age of the children was 4, 5, and 6 years, respectively.

### 4.2. Participant's Characteristics

#### 4.2.1. Sleep Characteristics

Sleep features of the SPATZ parents were collected using the Munich Chronotype Questionnaire (MCTQ), a validated tool used to determine sleep features in the general population [28]. The MCTQ was implemented from the 4-year follow-up on, so the present manuscript includes data from the follow-ups when the children were 4, 5, and 6 years old. In the present work, the following five sleeping features have been collected during free and working days indicated on average within the past 30 days: (1) Sleep duration was computed as the difference between wake-up time and sleep onset (SD); (2) Time spent in bed was computed as the difference between the time at which the subject gets out of bed and the time at which the subject goes to bed (TBT); (3) Midpoints of sleep was computed as the sum of sleep onset and half of the sleep duration (MS); (4) Local time of getting out of bed was computed as the sum of the time at which the sleep ended and the time the subject takes to get up after awakening (sleep inertia) (GUN); (5) Sleep Onset was defined as the sum of the time at which a subject goes to sleep and the time a subject takes to fall asleep (Sleep latency) (SON).

#### 4.2.2. Partner Relationship Assessment

The partner relationship assessment was performed employing the parent partner relationship subscale (PB) of the German version of the Parenting Stress Index ("Eltern-Belastungs-Inventar (EBI)"). Reliability and validity of PB have been shown in previous research [29]. In brief, the PB subscale is computed by the sum of four items evaluating relationship satisfaction of a couple since their child was born. Specifically, the PB subscale items are a five-level Likert scale assessing the following domains:

(1) Time spent with the partner; (2) Sexuality; (3) Leisure activities with the partner; (4) Problems in the relationship. The sum of the above items determines a score ranging between 5 and 20, with higher scores related to a lower partner relationship satisfaction.

### 4.2.3. Child Sleeping Quality

The total score from the German version of the Children's Sleep Habits Questionnaire (CSHQ) was used to determine the child's sleep quality during a typical week [30]. The CSHQ is a multidimensional tool in which parents were asked to provide information regarding bedtime resistance, sleep onset delay, sleep duration, sleep anxiety, night walking, parasomnias, sleep-disordered breathing, and daytime sleepiness. The CSHQ total score was computed by the sum of 34 items scored by a 3 level Likert scale, resulting in a score ranging between 34 and 102 with higher values indicating more sleep problems. Reliability and validity of CSHQ have been shown in previous research [31].

### 4.3. Statistical Methods

The median and interquartile range were used to describe sleep features and partner relationship scores. A within couple similarity index of sleep features and partner relationship score was calculated as a women-men difference [31] and reported by child age as a median and 10th–90th percentile range. The within couple comparison of women vs. men was performed by using an exact signed test based on the binomial distribution.

The correlation structure of couples' sleeping characteristics during working and free days and of partner relationship scores was reported by child age using a heat-map of Pearson correlation coefficients performed on Blom´s transformed variables [32]. A second set of heat maps was performed to show the time correlation structure of sleeping characteristics, the partner relationship score, and the children sleeping quality score in men and women, separately and during working and free days (Figures 2 and 3).

The Actor-Partner Interdependence Model (APIM) was used to investigate the association between the partner relationship score, sleeping features, and child sleep quality, as a whole. This paradigm was adopted to evaluate how a given feature of an individual may have influenced his/her own relationship (actor effect) and how this same feature may have influenced the relationship score reported by the partner (partner effect). In the present work, the APIM model was extended to take into account the child's sleep quality. To this end, we considered the direct effect of child sleep quality on the parent/mother relationship score along with its indirect effect through parental/maternal sleep.

The APIM analyses were performed considering the longitudinal design of the SPATZ study using a mixed model method based on a linear growth model [33]. In brief, a structural equation approach was used to model the latent factor of predictors over the latent factor of outcomes at the three time points. Model fitting was reported using the Bayesian Information Criteria (BIC). All models were adjusted for the baseline age of participants (continuous) and their education (a dichotomous indicator variable for a university degree). Blom´s transformed variables were used and standardized coefficients were reported.

Sensitivity analyses were conducted adjusting for parents' occupation (self-employee, leadership-managerial, and others), body mass index (continuous), parity (an indicator variable for more than one child), smoking status (indicator variable for ever-smoked), and alcohol use (indicator variable for regular users). In this second evaluation, indicator variables were used to take the missing data into account.

Structural equation modeling was conducted using the LAVAAN package of the R software (version 3.6). All statistical tests were two tailed and the type-I error rate was set to 5% ($\alpha = 0.05$).

## 5. Strength and Limitations

The current work has numerous strengths. Firstly, the current study is based on features collected using validated tools. In particular, the chronotype was defined using a wide spectrum of sleeping

features collected during working and free days. Secondly, our work was based on a rigorous and comprehensive statistical approach in which all the family members are considered as a whole, estimating model parameters using simultaneous regressions. Finally, the prospective design of the Ulm SPATZ Health Study allowed us to gather information on causal relationships and adopt a causal interpretation of the observed effects, with all the limits imposed by the nature of an observational study.

Further, it should be highlighted that sample size may have limited the current study, resulting in a possible lack of statistical power. This problem is relevant in the case of structural equation modeling applied to estimate numerous latent factors arising from weakly correlated variables with a sample size lower than 200 units. However, we believe that the type-II error was negligible in our study because of the observed strong correlation structures and because we applied a structural equation approach to a single latent factor model. Under this condition, simulations showed that the current sample size appears to be sufficient to guarantee a statistical power higher than 80% [34]. A second major methodological limitation could be residual confounding due to the use of a relatively simple models adjusted for few confounders. Here, the sequential modeling analysis performed indicated an acceptable level of homogeneity between the results from the simplest model and those from a much more complex model supplementary adjusted for occupation, body mass index, parity, smoking status, and alcohol use. Finally, the use of validated instruments does not completely avoid the possibility that our study was affected by a certain degree of measurement error. However, the current results accord with meaningful common sense results, suggesting internal consistency.

## 6. Conclusions

The present work confirms that sleep represents an essential part of the relationship satisfaction and stability of couples with young children. Moreover, we also showed how the sleeping features, the partnership satisfaction, and their correlation changes with child age. We here also reported how chronotype proxies and sleeping features are different correlated and influenced by working activities. In the present work, we showed how child sleep quality influences relationship satisfaction. Most importantly, we showed here that child sleep quality mostly influences relationship satisfaction in mothers. We interpret this to be likely because of the higher involvement of mothers in childcare during working days. However, its effect on the male partners is not negligible. Distress caused by a lack of good sleeping could then be counteracted by a major involvement of the fathers in looking after children during working days and by extensions of paternity leave from work, with parallel employment or a return to work of the mother. The current results are of an explorative nature and more accurate observational studies, based on objectively assessed evaluations of sleep, should be encouraged. Finally, the nature of our results was clearly influenced by local factors such as the mother's involvement in working activities and father's paternity role. The next steps in this field should be creating networks to study how sleeping influences wellbeing, quality of life, and partnership satisfaction on a global scale, by planning multicenter-international cohort studies.

**Author Contributions:** C.R. and J.G. conceived and conducted the study, C.R. performed the statistical analyses and wrote the first version of the manuscript, J.G. and D.R. are responsible and principal investigators of the SPATZ study. Z.P.-R. and J.F.D.-M. contributed to data interpretation. All authors have read and agreed to the published version of the manuscript.

**Funding:** This research was also partially funded by Programa Estatal de Investigación, Desarrollo e Innovación Orientada a Retos de la Sociedad (2017–2019). Ministerio de Economía y Competitividad, Spain [PSI2016-76552] to J.F.D.-M.

**Conflicts of Interest:** The authors declare no conflict of interest.

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
