# Peer review of "The Parent’s Chronotype and Child’s Sleeping Quality in Association with Relationship Satisfaction"

_2624-5175, doi:10.3390/clockssleep2030028_

Round 1
Reviewer 1 Report
- The literature review seems not comprehensive. Please add more recent journal publications.
- The paper needs to be formatted according to the template.
- Abbreviations should be explained the first time they appear in the paper.
- The Heat Map pictures need to be explained and discussed in more detail.
- The citation format needs to be modified according to the template.
- The conclusion section is very short and not comprehensive.
Author Response
1) The literature review seems not comprehensive. Please add more recent journal publications.
R: the literature Review has been revised and more papers have been cited
2) The paper needs to be formatted according to the template.
R: Formatting of template has been done by the Editor. However we will double check in further Phase of proofreading (in case the paper will be accepted)
3) Abbreviations should be explained the first time they appear in the paper.
Supplementary explanations to abbreviations have been added. Please consider that SPATZ (the Name of the study) has no abbreviations because it is the German word for "sparrow" the mascotte of the study
4) The Heat Map pictures need to be explained and discussed in more detail.
further explanations for the heat-maps have been addded
5) The citation format needs to be modified according to the template.
Citation Format have been revised according to Journal style
6) The conclusion section is very short and not comprehensive.
The conclusion have been extended
Reviewer 2 Report
1.Title – the second part can be removed. Also, please consider “Parents chronotype...” at the start of the title, as this would indicate triadic approach.
2.Abstract – Please enroll “SPATZ” abbreviation.
3.Introduction – “According to accepted theories, the couple and their child might be interpreted as a single unit” I guess it should be “single system” not “unit”. Anyway, can the authors provide a reference for these theories?
4.Methods (also abstract) – it is unclear who completed questionnaires. It appears that the respondents were only mothers who provided responses for themselves, their partners, and their children. If this was the case it must be clearly stated and considered in limitations, if not, it also must be clarified.
5.Results (also abstract) – “211 couples remained in the analytical dataset” – couples or triads? Please be consistent throughout the manuscript.
6.Discussion - Associations of different sleep timing characteristics with relationship satisfaction are interesting, but are they limited to this specific type of satisfaction? The authors state in conclusions that “The present work confirms that sleep represents an essential part of the wellbeing”. I would like the authors to discuss this topic in reference to previous studies, both cross-sectional and longitudinal, to show more general pattern.
Author Response
1.Title – the second part can be removed. Also, please consider “Parents chronotype...” at the start of the title, as this would indicate triadic approach.
R: The title was changed to "Parent’s chronotype and child sleeping quality in association with relationship satisfaction"
2.Abstract – Please enroll “SPATZ” abbreviation.
R: SPATZ is the german word for "sparrow" the Symbol of the study. SPATZ then is nor an acronym and does not have an abbreviation
3.Introduction – “According to accepted theories, the couple and their child might be interpreted as a single unit” I guess it should be “single system” not “unit”. Anyway, can the authors provide a reference for these theories?
R: this sentence was revised
4.Methods (also abstract) – it is unclear who completed questionnaires. It appears that the respondents were only mothers who provided responses for themselves, their partners, and their children. If this was the case it must be clearly stated and considered in limitations, if not, it also must be clarified.
R: this was clarified in the current version, basically both parents participated responding the questionnaires
5.Results (also abstract) – “211 couples remained in the analytical dataset” – couples or triads? Please be consistent throughout the manuscript.
R: The term triads replaced couple. In the titles of tables 1-2 we kept couple because the data refers to men and women
6.Discussion - Associations of different sleep timing characteristics with relationship satisfaction are interesting, but are they limited to this specific type of satisfaction? The authors state in conclusions that “The present work confirms that sleep represents an essential part of the wellbeing”. I would like the authors to discuss this topic in reference to previous studies, both cross-sectional and longitudinal, to show more general pattern.
R: we revised the sentence avoiding the General term "wellbeing" and we substituted it with relationship satisfaction. The conclusion have been also Extended. Being limited to this specific aspect of the partnership satisfaction and considering the scarcity of available literature we preferred to not extend the current discussion to avoid possible speculations
Round 2
Reviewer 1 Report
The captions for figures are wrong. It needs to be corrected.
I still feel that the first three figures need to be accompanied with further analysis and discussion.
For the chosen format for figure 4, it will help if the authors explain one of the cases and show how the figure and data can be interpreted.
Author Response
The captions for figures are wrong. It needs to be corrected.
R: We checked and the current Version has right captures for figures.
I still feel that the first three figures need to be accompanied with further analysis and discussion.
R: We agree that we could have improved this part. However, those simple univariate correlation (here represented by the correlogram or heatmaps) are generally of easy Interpretation for the Reader. We then preferred to Focus on the more comprehensive 3D complex of APIM. Notably, the APIM model here, along with similar metric for the Regression slopes (as defined as correlation coefficients) represent an implementation of the simple univariate correlations of figures 1-3 and also represent a further Analysis of the correlations.
For the chosen format for figure 4, it will help if the authors explain one of the cases and show how the figure and data can be interpreted.
We added a supplementary part starting from line 210 to 218
Reviewer 2 Report
thanks
Author Response
It was a pleasure to follow your valuable suggestions and we sincerely thank you for the contribution